# Association between prostate cancer characteristics and *BRCA1/2*-associated family cancer history in a Japanese cohort

Yudai Ishiyama[1,2]*, Masaki Shimbo[1], Junpei Iizuka[2], Gautam Deshpande[3,4], Kazunari Tanabe[2], Kazunori Hattori[1]

**1** Department of Urology, St. Luke's International Hospital, Tokyo, Japan, **2** Department of Urology, Tokyo Women's Medical University, Tokyo, Japan, **3** Support Unit for Conducting Clinically Essential Study, St. Luke's International University, Tokyo, Japan, **4** Department of General Internal Medicine, St. Luke's International Hospital, Tokyo, Japan

* yuishi0831@gmail.com

**Data Availability Statement:** The data contain family history data, which are considered sensitive

## Abstract

In addition to breast, ovarian, and pancreatic cancers, *BRCA1/2* genes have been associated with prostate cancer (PC). However, the role of *BRCA1/2*-associated family cancer history (FCH) has remained unexplored in treating these four cancer types as a homogenous pathophysiological group. We aimed to clarify the relationship between *BRCA1/2*-associated FCH and PC, and to assess its relationship with cancer aggressiveness. Patient characteristics, positive family history of *BRCA1/2*-associated cancer, and cancer characteristics (Gleason score, prostate specific antigen level at diagnosis, and clinical tumor stage) were analyzed. Among the 1,985 eligible candidates, 473 (23.83%) patients had adequately detailed FCH, obtained via questionnaire, and were thus included in the study. *BRCA1/2*-associated FCH was observed in 135 (28.54%) patients with PC (68, 14.38%), breast (44, 9.30%), pancreatic (31, 6.55%), or ovarian (8, 1.69%) cancers. *BRCA1/2*-associated FCH was not significantly associated with high Gleason score ($\geq$ 8). Patients with *BRCA*-associated FCH were less likely to present with high clinical tumor stage, and no difference was observed in prostate-specific antigen level, presence of metastatic lesions at diagnosis, or likelihood of high-risk classification between patients with and without *BRCA*-associated FCH. This is the first report of *BRCA1/2*-associated FCH in Japanese men, indicating that family history did not affect the severity or aggressiveness of PC.

## Introduction

Prostate cancer (PC) is the third leading malignancy among Japanese men, and is suggested to have a strong genetic component [1]. Accordingly, hereditary risk for PC has been defined as 1) more than three first-degree relatives with PC, 2) three successive generations with PC, or 3) two relatives affected under 55 years of age [2]. Numerous trials have been conducted to explore PC-associated genes, leading to a consensus that PC is a highly heterogeneous malignancy [3]. *BRCA1/2* genes, originally known for their association with breast (BC) and ovarian

identifying information by Institutional Review Board of St. Luke's International Hospital (Tokyo, Japan) based on current Japanese Act on Protection of Personal Information. Additionally, this study was conducted with opt-out recruitment, so the Institutional Review Board does not allow for the minimal data set to be made publicly available. Any reader or researcher can contact Research Management Office, St. Luke's International Hospital, Tokyo, Japan (contact via kenkyukikaku@luke.ac.jp) for request to access the dataset.

**Funding:** The authors received no specific funding for this work.

**Competing interests:** The authors have declared that no competing interests exist.

(OvC) cancers, has also been implicated in PC and pancreatic cancer (PaC) [4]. While *BRCA2* seems to play a greater role in PC than *BRCA1*, differentiation between the two in a clinical setting is relatively difficult, with both genes being used as inclusion criteria for clinical trials evaluating the efficacy of PC treatments [5]. Previous studies have shown that a family history of PC and BC is associated with PC risk [6–8]; however, the association between *BRCA1/2*-related family cancer history (FCH) and PC aggressiveness remains controversial [9–12]. While the relationship between PC, *BRCA1/2*, and FCH represents a field of interest for clinicians, only a few studies have explored this potential association in Japanese populations [9,13], and no study has focused specifically on PC severity and *BRCA1/2*-associated FCH in this population. Performing genetic aberration analyses in Japan are challenging due to both cost and accessibility issues, and it remains unclear whether consideration of FCH actually provides additional pertinent information or simply directs genetic testing. Further genetic testing and risk stratification would be facilitated by the identification of a relationship between FCH and PC aggressiveness.

In this context, the current study aimed to clarify *BRCA1/2*-associated FCH in Japanese patients with PC, identify family cohorts, and assess the relationship between *BRCA1/2*-associated FCH and PC aggressiveness.

## Patients and methods

### Patients and data collection

This was a retrospective, cross-sectional study. A total of 2286 patients with biopsy-confirmed PC, treated at St. Luke's International Hospital (Tokyo, Japan) between July 2003 and September 2017, were identified via electronic medical chart review. Those with insufficient clinical data were excluded, the remaining candidates were eligible for FCH querying. A family history questionnaire (S1 Fig) was provided to each patient at a follow-up clinical visit between May and September 2017. As the follow-up protocol for PC at our institution involves clinic visits at least every three months, we assumed full access to all eligible patients, as long as they remained paneled at our hospital. Only those patients with confirmed FCH data on the standardized questionnaire were eligible for inclusion in the analysis (Fig 1). PC, BC, OvC, and PaC were considered *BRCA1/2*-associated cancers. Positive *BRCA1/2*-associated FCH was defined as a single second-degree family member with such cancers, as per the recommendations of the National Comprehensive Cancer Network (NCCN) suggesting genetic testing for PC patients with a strong family history of PC and multiple ($\geq$3) cancers of a particular type, and NCCN guidelines for genetic/familial high-risk assessment of PaC, BC, and OvC [14,15]. The FCH data included: positive *BRCA1/2*-associated FCH, details of FCH, presence of any other FCH, and primary treatments received. FCH data was collected up to all second-degree relatives. Additional clinical data for eligible patients were extracted from their electronic charts. The study was conducted in accordance with the ethical standards of the Internal Ethics Review Board of St. Luke's International Hospital (approval number 17-J017) and with the 1964 Declaration of Helsinki. All patients provided written informed consent for treatment and collection of their data.

### Study protocol and outcomes

First, associations were determined between our independent variable, FCH (recorded as any *BRCA1/2*-associated cancer, as well as by PC, BrC, PaC, and OvC cancer type), and the primary dependent variable, high ($\geq$ 8) Gleason score (GS). Covariates in this analysis included: age at diagnosis, body mass index (BMI), smoking history, prostate specific antigen (PSA) level at diagnosis, high clinical tumor (cT) stage defined as $\geq$ cT3a per NCCN Practice

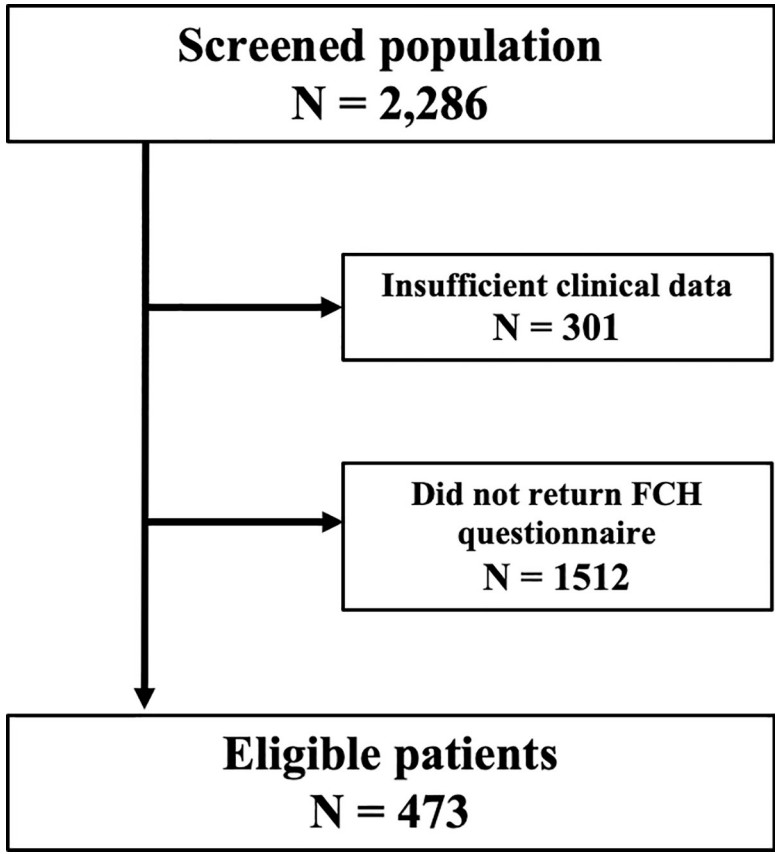

**Fig 1. Flowchart illustrating the study design.** FCH, family cancer history.

Guidelines in Oncology (Ver 2.2017), and the presence of other FCH not included in the dependent variable above. Next, patients were classified into two groups: those with and without *BRCA1/2*-associated FCH. We then compared the following dependent variables: high GS, PSA level at diagnosis (median), and high cT stage. Covariates included: age at diagnosis, BMI, smoking history, and presence of other FCH based on previously reported factors shown to influence PC severity or survival [16–18]. We retrospectively consulted patient electronic medical charts and MRI images to verify if cT staging was appropriate. In questionable cases, a decision was made by two clinicians (radiologist and urologist) before being included in the analysis. GS was assigned based on the results reported by a team of pathologists at a single institution.

Additionally, candidates were identified with high risk for *BRCA1/2* mutations. This was achieved by modifying the original hereditary risk factors for PC, defined by Carter et al., into a novel risk factor for *BRCA1/2*-associated cancer [2]. This new definition included > three first-degree relatives affected by *BRCA1/2*-associated cancer (group A); three successive generations of *BRCA1/2*-associated FCH (group B); and two relatives < 55 years of age, within the second degree, diagnosed with *BRCA1/2*-associated cancer (group C). The number of patients included in each group, as well as those who satisfied multiple criteria, was assessed.

## Statistical analysis

Univariate analysis was first performed, followed by multivariate logistic regression. Continuous variables were analyzed using Mann–Whitney *U*-test and categorical variables were

analyzed using the $\chi^2$ test or Fisher's exact test, with 95% confidence interval (CI). All analyses were performed using JMP software (Ver. 14.0; SAS Institute, Cary, NC, USA), and differences were considered statistically significant at p-values of $< 0.05$.

## Results

### Patient characteristics

Among 1,985 candidates eligible for FCH data collection, 473 patients (23.83%) returned completed FCH questionnaires and were included in the main analysis. These patients were generally diagnosed with PC later in the study period (median year of diagnosis, 2015), with $> 75\%$ being diagnosed after 2010, compared with the remaining 1,512 candidates in the study who did not respond to the questionnaire (median year of diagnosis, 2007). Patient characteristics of both cohorts are described in Table 1. As information regarding cT staging and the presence of metastatic lesions required descriptive chart review, which was only approved by the Internal Ethics Review Board for patients meeting inclusion criteria, this information is not presented for non-questionnaire responders (remaining patients).

Among the 473 patients included for analysis, the mean age was 67.0 years (IQR = 63.0–72.0) and the median serum PSA level at diagnosis was 7.14 ng/mL (IQR = 5.15–11.14). Among these, 180 patients (38.0%) had GS $\geq 8$, of which 338 (71.5%) underwent radical prostatectomy, and 9 (1.9%) had metastasis at diagnosis.

### Description of FCH

*BRCA1/2*-associated FCH of any type was identified in 135 (28.5%) patients. FCH of PC, BC, PaC, and OvC was found in 68 (14.3%), 44 (9.3%), 31 (6.6%), and 8 (1.7%) patients, respectively. For cancer types not included in our *BRCA1/2*-associated FCH criteria, the most common malignancies were gastric, colorectal, and lung cancers, observed in 49 (10.3%), 30 (6.3%), and 28 (5.9%) patients, respectively. One patient reported an FCH of both BC and OvC.

**Table 1. Patient characteristics.**

|  | Analyzed cohort N = 473 | Remaining patients N = 1,512 | P value |
|---|---|---|---|
| **Age at diagnosis, median (IQR)** | 67.00 (63.00–72.00) | 69.00 (63.00–74.00) | 0.009 |
| **BMI, median (IQR)** | 23.66 (21.94–25.49) | 23.48 (21.67–25.14) | 0.103 |
| **History of smoking (%)** | 248 (52.43) | 980 (64.82) | <0.001 |
| **Radical prostatectomy (%)** | 338 (71.46) | 880 (58.20) | <0.001 |
| **PSA at diagnosis (ng/mL), median (IQR)** | 7.14 (5.15–11.14) | 4.90 (2.22–9.37) | <0.001 |
| **Gleason score $\geq 8$ (%)** | 180 (38.05) | 534 (35.32) | 0.445 |
| **Gleason score $\geq 9$ (%)** | 92 (19.45) | 248 (16.40) | 0.794 |
| **Clinical T stage** |  |  |  |
| **$\leq$ T2a (%)** | 323 (68.29) | N/A |  |
| **T2b (%)** | 37 (7.82) | N/A |  |
| **T2c (%)** | 42 (8.88) | N/A |  |
| **$\geq$ T3 (%)** | 69 (14.59) | N/A |  |
| **Unknown (%)** | 2 (0.42) | N/A |  |
| **Metastasis at diagnosis (%)** | 9 (1.90) | N/A |  |

Gleason score (GS) $\geq 8$ also included patients with GS $\geq 9$. Clinical T stage was defined per TNM classification. BMI, body mass index; IQR, interquartile range; N/A, not applicable; PSA, prostate specific antigen.

## Association between *BRCA1/2*-associated FCH and cancer severity

Univariate analysis (Table 2) did not demonstrate a significant association between *BRCA1/2*-associated FCH and high GS ($\geq 8$) (OR: 0.94 [95% CI: 0.62–1.42], p = 0.773, p = 0.773). Moreover, no individual *BRCA1/2*-associated cancer type demonstrated an association with high GS; PC (OR: 1.17 [95% CI: 0.69–1.97], p = 0.567), BrC (OR: 0.92 [95% CI: 0.48–1.76], p = 0.808), PaC (OR: 0.55 [95% CI: 0.24–1.25], p = 0.146), and OvC (OR: 0.54 [95% CI: 0.11–2.69], p = 0.443). In the multivariate logistic regression model (Table 2), *BRCA1/2*-associated FCH did not show a significant association with high GS (OR: 0.83 [95% CI: 0.53–1.32], p = 0.773). Additionally, no specific *BRCA1/2*-associated cancer type demonstrated a significant association with high GS (S1–S4 Tables). Patients with a family history of multiple cancer types, such as PC and BrC, were too few to warrant statistical analysis. The PSA level was significantly higher among patients with high GS ($\geq 8$) than among patients with low GS ($< 8$) (median 9.20 vs. 6.30, p = 0.001).

## Association between *BRCA1/2*-associated FCH and other outcomes

Univariate analysis determined that patients with *BRCA1/2*-associated FCH were significantly less likely to present with high cT stage (OR = 0.45 [95% CI: 0.16–0.87], p = 0.017). In addition to having no direct association with GS $\geq 8$, *BRCA1/2*-associated FCH was not associated with PSA level (median 7.20 [IQR = 5.03–10.95] vs. 7.13 [5.17–11.29]) or the presence of metastatic lesions at diagnosis (OR = 0.71 [95% CI: 0.15–3.47], p = 0.664). Meanwhile, significantly higher BMI values were observed among patients with *BRCA1/2*-associated FCH (median 23.95 vs 23.47, p = 0.039). There was only marginal correlation between positive *BRCA1/2*-associated FCH and other FCH (OR = 1.45 [95% CI: 0.91–2.29], p = 0.069). Similar results were obtained by multivariate analysis, with *BRCA1/2*-associated FCH independently associated with BMI (p = 0.023) and less so with high cT stage (p = 0.010). These results are summarized in Table 3.

## Association between *BRCA1/2* FCH and other outcomes in localized cancer

We performed subgroup analyses of patients with localized PC (N = 467). Similar to the overall population, patients with *BRCA1/2*-associated FCH were significantly less likely to present

**Table 2. Association between GS and variables including *BRCA1/2*-associated FCH.**

|  | GS < 8 | GS ≥ 8 | Odds ratio ≥ 8 / < 8 [95% CI] | Univariate p-value | Multivariate p-value |
|---|---|---|---|---|---|
| **Age, median (IQR)** | 67 (63–72) | 68 (63–74) | - | 0.106 | 0.530 |
| **BMI, median (IQR)** | 23.66 (21.97–25.36) | 23.52 (21.81–25.54) | - | 0.886 | 0.737 |
| **History of smoking (%)** | 158 (56.83) | 90 (54.55) | 0.94 [0.62–1.43] | 0.639 | 0.588 |
| **PSA, median (IQR)** | 6.30 (4.98–8.77) | 9.20 (5.86–20.5) |  | < 0.001 | < 0.001 |
| **High clinical T stage** | 18 (6.14) | 49 (27.34%) | 3.84 [1.97–7.48] | < 0.001 | < 0.001 |
| ***BRCA1/2* FCH** | 85 (29.01) | 50 (27.78) | 0.83 [0.53–1.32] | 0.773 | 0.743 |
| **Prostate Cancer** | 40 (13.65) | 28 (15.56) | 1.36 [0.76–2.42] | 0.567 |  |
| **Breast Cancer** | 28 (9.56) | 16 (8.89) | 0.87 [0.44–1.75] | 0.808 |  |
| **Pancreatic Cancer** | 23 (7.85) | 8 (4.44) | 0.77 [0.32–1.82] | 0.146 |  |
| **Ovarian Cancer** | 6 (2.05) | 2 (1.11) | 0.58 [0.11–3.15] | 0.443 |  |
| **Other FCH** | 85 (29.01) | 48 (26.67) | 1.06 [0.67–1.61] | 0.582 | 0.952 |

FCH of prostate, breast, pancreatic, and ovarian cancer in individuals was not included in multivariate analysis of this table. See S1–S4 Tables for these results. Clinical T stage was defined per TNM classification. BMI, body mass index; CI, confidence interval; FCH, family cancer history; GS, Gleason score; IQR, interquartile range; PSA, prostate specific antigen. (N = 473)

**Table 3. Association between *BRCA1/2*-associated FCH and variables in all patients (N = 473).**

|  | *BRCA1/2* FCH - | *BRCA1/2* FCH + | Odds ratio +/- [95% CI] | Univariate p-value | Multivariate p-value |
|---|---|---|---|---|---|
| **Age, median (IQR)** | 67.51 (63.00–73.00) | 66.00 (62.50–72.00) | - | 0.184 | 0.429 |
| **BMI, median (IQR)** | 23.48 (21.88–25.26) | 23.95 (22.43–26.05) | - | 0.039 | 0.023 |
| **History of smoking (%)** | 175 (56.60%) | 71 (56.80%) | 0.975 [0.61–1.43] | 0.893 | 0.756 |
| **PSA, median (IQR)** | 7.13 (5.17–11.29) | 7.20 (5.03–10.95) | - | 0.558 | 0.510 |
| **High clinical T stage** | 56 (16.6%) | 11 (8.2%) | 0.45 [0.16–0.87] | 0.017 | 0.010 |
| **GS ≥ 8** | 130 (38.12%) | 50 (37.04%) | 0.95 [0.63–1.44] | 0.826 | 0.744 |
| **Other FCH** | 87 (25.7%) | 46 (34.1%) | 1.45 [0.91–2.29] | 0.069 | 0.087 |
| **Metastasis at diagnosis** | 7 (20.7%) | 2 (1.48%) | 0.71 [0.15–3.47] | 0.664 | 0.992 |

Clinical T stage was defined per TNM classification. BMI, body mass index; CI, confidence interval; FCH, family cancer history; GS, Gleason score; IQR, interquartile range; PSA, prostate specific antigen.

with high cT stage (OR = 0.45 [95% CI: 0.22–0.92], p = 0.019) and had significantly higher BMI values (median 23.49 [IQR: 21.86–25.29] vs. 23.95 [22.48–26.04], p = 0.041). Furthermore, no direct association was observed between *BRCA1/2*-associated FCH and high GS (OR = 0.94, [95% CI: 0.62–1.43], p = 0.787) or NCCN high-risk classification (OR = 0.81 [95% CI: 0.54–1.22], p = 0.315) following univariate analysis. Similarly, multivariate analysis demonstrated that *BRCA1/2*-associated FCH was independently associated with BMI (p = 0.022) and to a lesser degree with high cT stage (p = 0.048), but was not associated with high GS (p = 0.499) or NCCN high-risk classification (p = 0.525). These results are summarized in Table 4.

## Identification of high-risk candidates

To identify candidates with high potential for *BRCA1/2* mutations, we performed further analyses by modifying the original hereditary risk factors for PC into a novel risk factor for *BRCA1/2*-associated cancer [2]. Twenty-four (5.07%) patients had more than three first-degree relatives affected by *BRCA1/2*-associated cancer (group A), two (0.42%) patients had three successive generations of *BRCA1/2*-associated FCH (group B), and five (1.06%) patients had two relatives diagnosed < 55 years of age (group C). One (0.21%) and three (0.63%) patients met criteria for both groups A/B and C/A, respectively, and 26 (5.50%) patients met at least one of three criteria. No patient met all criteria (Fig 2). Genetic testing was not performed in this study.

**Table 4. Association between *BRCA1/2*-associated FCH and variables in localized cancer (N = 467).**

|  | *BRCA1/2* FCH–(N = 133) | *BRCA1/2* FCH + (N = 331) | Odds ratio +/- [95% CI] | Univariate p-value | Multivariate p-value |
|---|---|---|---|---|---|
| **Age, median (IQR)** | 67.00 (63.00–73.00) | 66.00 (62.50–72.00) | – | 0.200 | 0.398 |
| **BMI, median (IQR)** | 23.49 (21.86–25.29) | 23.95 (22.48–26.04) | – | 0.041 | 0.022 |
| **History of smoking (%)** | 175 (52.87%) | 71 (53.38%) | 1.02 [0.68–1.53] | 0.920 | 0.767 |
| **PSA, median (IQR)** | 7.10 (5.15–11.10) | 7.14 (5.00–10.52) | – | 0.825 | 0.656 |
| **NCCN high-risk** | 145 (43.41%) | 51 (38.35%) | 0.81 [0.54–1.22] | 0.315 | 0.526 |
| **High cT stage** | 51 (15.27%) | 10 (7.52%) | 0.45 [0.22–0.92] | 0.019 | 0.048 |
| **GS ≥ 8** | 125 (37.43%) | 48 (36.09%) | 0.94 [0.62–1.43] | 0.787 | 0.499 |
| **Other FCH** | 86 (25.75%) | 46 (34.59%) | 1.51 [0.98–2.32] | 0.059 | 0.093 |

Clinical T stage was defined per TNM classification. BMI, body mass index; CI, confidence interval; FCH, family cancer history; GS, Gleason score; IQR, interquartile range; NCCN, National Comprehensive Cancer Network; PSA, prostate specific antigen.

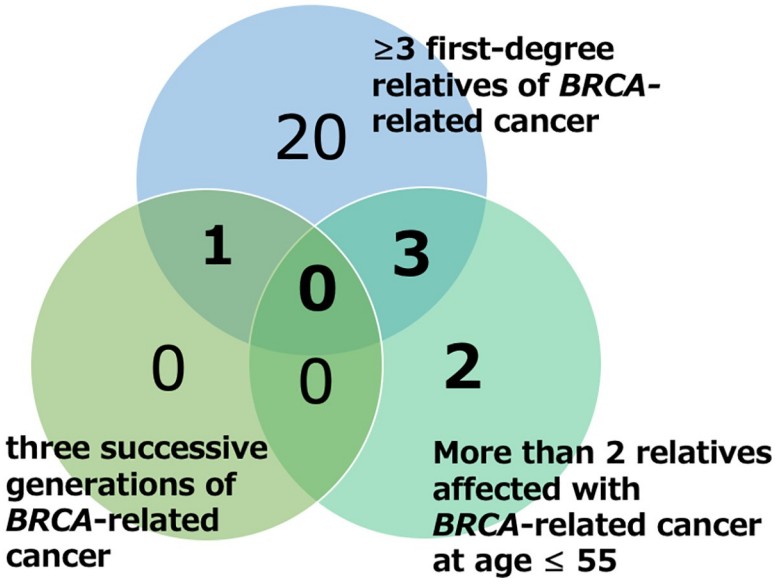

**Fig 2. The proposed novel risk factor for *BRCA1/2*-associated cancer.**

## Discussion

In the current study, no direct association was observed between *BRCA1/2*-associated FCH and GS. Similarly, subgroup analyses of patients with localized cancer demonstrated that *BRCA1/2*-associated FCH was not associated with NCCN high-risk classification. Patients with *BRCA1/2*-associated FCH were less likely to present with high T stage.

The association between FCH and PC severity remains controversial, for PC as well as other *BRCA1/2*-associated cancers. One large-scale study from Sweden suggested concordance of survival in family members with PC [11]. If so, combining family histories of cancers of different severity, as in this study, might have contributed to our lack of significant findings. However, it is unlikely that a positive family history would directly lead to any change in disease management. We found that *BRCA1/2*-associated FCH was potentially related to early-stage cancer. However, no other clinical marker, such as PSA level, demonstrated a significantly positive association with *BRCA1/2*-associated FCH. A previous study of 2,019 patients with PC and known *BRCA1/2* germline mutations reported an association with higher risk of nodal involvement, distant metastasis, and poor overall survival [19]. Another study in a similar population suggested that treatment response after radical prostatectomy may be worse in patients with *BRCA1/2* mutations [20]. Although it was difficult to estimate how many of our FCH-positive patients harbored *BRCA1/2* germline mutations, it seems unlikely that a *BRCA1/2*-associated FCH would contribute to improved prognosis. Thus, we presume that patients with a positive FCH may have greater disease awareness, interest in screening, and, consequently, undergo earlier detection. If so, estimating PC severity or aggressiveness based on FCH information alone may be challenging. Hence, caution is warranted for protocol planning in future studies.

To the best of our knowledge, this is the first study to report complete *BRCA1/2*-associated FCH of patients with PC in the Japanese population. By including family histories of any *BRCA1/2*-associated cancer, the FCH population of our study almost doubled compared to that of PC only. Interestingly, 6.6% of our study population had an FCH of PaC, which was larger than in previous reports [21]. As this is an observational study, disease prevalence could not be assessed. In Japan, there is currently no available nationwide registry database, making it challenging to obtain exhaustive family history data. Therefore, the data for this study may prove beneficial to physicians providing care for patients with PC, not just within Japan, but across Asia. Results from previous studies suggested that *BRCA1* and *BRCA2* may behave differently in the context of PC, and that *BRCA2* has greater impact on PC risk [22,23]. However, differentiation between *BRCA1* and *BRCA2* is relatively difficult in a clinical setting, resulting in both being used for inclusion criteria of several clinical trials for metastatic PC [5,24]. Thus, treating the two gene mutations as a single group remains valid in light of current practice.

We also sought to identify probable candidates for the *BRCA1/2* family cohort as no direct association was observed within the general population. We originally estimated the rate of *BRCA1/2*-associated PC based on data from the *BRCA1/2*-associated BrC population and *BRCA1/2* mutation analysis of the PC population. Castro et al. reported that 3.89% of PC patients < 65 years old or with a first-degree FCH of PC had *BRCA1/2* mutations (*BRCA1*: 0.89%, *BRCA2*: 3.0%) [19]. Maier et al. estimated 1.0% of patients with PC harbored *BRCA2* [25]. These results implied that roughly 1–4% of all PC patients possess *BRCA1/2* mutations, which is substantially lower than the 5.50% of patients included in one of our novel risk factor groups (A, B, or C). However, as we were unable to conduct confirmatory genetic testing in this study, the accuracy of our risk factor model warrants further validation.

Obtaining concrete FCH data offers several possible benefits. Most obviously, it is the initial step required to obtain somatic genetic testing, recommended for at-risk patients [14,15]. However, performing genetic aberration analysis is currently challenging in Japan, due to both high costs as well as accessibility to genetic counseling, resources for which remain extremely limited in Japan. Screening of patients using FCH is easy and inexpensive when performed accurately. Another meaningful aspect of grouping patients according to FCH is to increase sensitivity. Indeed, family cohorts of *BRCA1/2* germline mutations have been recognized in patients with BrC. However, it may become increasingly difficult to obtain detailed FCH information across several generations due to temporal trends toward nuclear families. Approximately 30% of patients with PC possess some degree of *BRCA1/2*-associated FCH, making FCH information valuable. Establishing FCH cohorts for several different cancer types may encourage the pursuit of earlier screening, leading to even earlier detection.

Certain limitations were noted in this study. First, we did not perform *BRCA1/2* gene detection due to both ethical and cost concerns in Japan. Additionally, since the questionnaire was administered to patients at follow-up clinic visits, the time between diagnosis and completing the questionnaire was not standardized between patients. Moreover, although we asked virtually every follow-up patient in our clinic to participate, only 20% fulfilled our inclusion criteria due to the short data collection time. This was reflected in the distribution of the year of PC diagnosis in the responding cohort versus the remaining patients. Additionally, the majority of patients were relatively early in their post-operative course, and FCHs were excluded from elderly patients treated long before the study period. These factors might raise concern for selection bias. Indeed, 38% of patients had GS $\geq$ 8, higher than previously reported for Japanese PC patients [26]. Thus, to what extent our cohort precisely represents the current Japanese PC population remains unclear. Mateo et al. analyzed genomic aberrations in 50 patients with metastatic castration-resistant PC (CRPC), identifying seven (14%) with *BRCA2* aberrations [5]. However, only three (6%) had germline mutations, even in the CRPC cohort. The

population in our study had rather early-stage cancer. Hence, if patients had been included with metastatic PC or CRPC, a different outcome may have been reached. Moreover, the sample size was small, hampering detection of statistical significance, especially for the small percentage of *BRCA1/2*-associated patients. Further, the criteria for grading GS were revised in 2014 by the International Society of Urological Pathology, and relevant changes in a pathologist's perception toward classification might have resulted in an undetermined effect [27]. Finally, as is always the case with collecting FCH and considering that most patients with PC were elderly, the existence of recall bias cannot be overlooked.

## Conclusions

In conclusion, this is the first descriptive report of complete *BRCA1/2*-associated FCH among patients with PC in Japan. *BRCA1/2*-associated FCH was not associated with severity or aggressiveness of PC. Further investigation of this population and FCH of related cancers may uncover candidates with high potential for *BRCA1/2* mutations.

## Supporting information

**S1 Fig. English translation of questionnaire used.**
(DOCX)

**S1 Table. Association between GS and variables including family history of prostate cancer.**
(DOCX)

**S2 Table. Association between GS and variables including family history of breast cancer.**
(DOCX)

**S3 Table. Association between GS and variables including family history of pancreatic cancer.**
(DOCX)

**S4 Table. Association between GS and variables including family history of ovarian cancer.**
(DOCX)

## Acknowledgments

We thank Dr. Junko Takei (Breast Surgical Oncology, St. Luke's International Hospital) and Misato Suzuki (Department of Clinical Genetics, St. Luke's International Hospital) for their advice regarding the study concept and general background.

## Author Contributions

**Conceptualization:** Yudai Ishiyama, Masaki Shimbo.

**Data curation:** Yudai Ishiyama, Masaki Shimbo.

**Funding acquisition:** Masaki Shimbo.

**Investigation:** Yudai Ishiyama.

**Methodology:** Yudai Ishiyama, Gautam Deshpande.

**Resources:** Yudai Ishiyama.

**Supervision:** Masaki Shimbo, Junpei Iizuka, Gautam Deshpande, Kazunari Tanabe, Kazunori Hattori.

**Writing – original draft:** Yudai Ishiyama.

**Writing – review & editing:** Yudai Ishiyama, Masaki Shimbo, Junpei Iizuka.

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
