## [Decision Letter · Decision Letter 0]

21 Aug 2020

PONE-D-20-24272

Association between prostate cancer characteristics and BRCA-associated family cancer history in a Japanese cohort

PLOS ONE

Dear Dr. Ishiyama,

Thank you for submitting your manuscript to PLOS ONE. After careful consideration, we feel that it has merit but does not fully meet PLOS ONE’s publication criteria as it currently stands. Therefore, we invite you to submit a revised version of the manuscript that addresses the points raised during the review process.

The reviewers have identified some substantial concerns that need to be addressed.  These include:

1.  Potential for bias in ascertainment of participants (unusually high overall Gleason scores) as well as in follow-up.  It is of critical importance to address specific concerns about study bias raised by the reviewers.

2.  The definition of "high-risk" being based entirely on Gleason score.  See reviewer's comments on other factors to include.

3.  There are already clinical recommendations in the United States that suggest that men with high risk prostate tumors undergo genetic testing for BRCA2 as they are tightly correlated.  In the introduction and background these studies, as well as the NCCN guidelines, should be references and results here should be put into the context of these data.

4.  BRCA1 and BRCA2 appear to be considered similarly.  Existing evidence suggests that BRCA2 is much more likely linked to aggressive prostate cancer (and prostate cancer in general) compared to BRCA1.  Re-assess your data in this context and update your background and discussion to better highlight these differences.

5.  More details on how families were selected for inclusion and details on the factors to be considered in the multivariate model.

6.  Address the other points raised by the reviewers.

We look forward to receiving your revised manuscript.

Kind regards,

Amanda Ewart Toland, Ph.D.

Academic Editor

PLOS ONE

Journal Requirements:

2. Please ensure that you refer to Figure 1 in your text as, if accepted, production will need this reference to link the reader to the figure.

3. Your ethics statement must appear in the Methods section of your manuscript.

If your ethics statement is written in any section besides the Methods, please move it to the Methods section and delete it from any other section.

Please also ensure that your ethics statement is included in your manuscript, as the ethics section of your online submission will not be published alongside your manuscript.

Reviewers' comments:

Reviewer's Responses to Questions

**Comments to the Author**

1. Is the manuscript technically sound, and do the data support the conclusions?

Reviewer #1: Partly

Reviewer #2: Partly

2. Has the statistical analysis been performed appropriately and rigorously? 

Reviewer #1: No

Reviewer #2: Yes

3. Have the authors made all data underlying the findings in their manuscript fully available?

Reviewer #1: Yes

Reviewer #2: Yes

4. Is the manuscript presented in an intelligible fashion and written in standard English?

Reviewer #1: Yes

Reviewer #2: Yes

5. Review Comments to the Author

Reviewer #1: This is a retrospective cross sectional study of Japanese men with prostate cancer, and whether presence of BRCA-associated family history is associated with aggressiveness of prostate cancer based on PSA, Gleason Grade, and T stage at diagnosis. They did not find any association between BRCA fam history and cancer aggressiveness. The topic is novel & timely, and the manuscript is overall well written aside from minor language errors. However, several methodological flaws limit validity of findings.

Major comments

- The authors refer to "BRCA genes," but BRCA is a family of genes that includes BRCA1 and BRCA2. The distinction is important as BRCA1 variant carriers have only a slightly greater risk of prostate cancer than non-carriers, versus BRCA2 where the risk of prostate cancer is 7-fold. This may have implications on prostate cancer aggressiveness. Though it is difficult to differentiate a BRCA1-family history vs BRCA2-family history, the authors should nevertheless address this.

- It is difficult to grasp the main message in the Discussion.

- The cohort-defining variable of "BRCA-associated FCH" is not defined in the Methods. Specifically, which cancers and confirming Carter et al wa sused.

- The primary endpoint of Gleason score is not justified. In the USA, a composite "aggressiveness" score including Gleason, PSA, and T stage are combined to create a Grade Group which is used to make management decisions - never one alone. Similarly, it is unclear why regional and metastatic disease are not included as "aggressive" prostate cancer.

- There is high potential for selection bias. It is not described in the methods to whom/how/when the questionnaire that ultimately defines family history is sent. Only 20% of patients were eligible and evaluable for the outcome, but this is not quantified in Results. On my review, it does appear that there is bias: 57% of patients had a Gleason score of 8 or higher, which is incredibly high. The finding of association of BRCA Fam History with *early* tumor stage is inconsistent with prior studies of positive known BRCA variant carriers as the authors mention, which also raises the question of bias. The authors admit that it is challenging to estimate aggressive based on family history information only. Moreover, the lack of BRCA mutation status in the study cohort is a major limitation.

- It is not described how the variables in the multivariate model are selected. Table 2 includes 11 variables, some of which are co-linear (e.g. BRCA FCH and Prostate) which does not seem appropriate. Why is smoking status included in the model?

- Several analyses are not described in the methods. For example, Table 3 is reported but unclear what this adds to the message. Tables 4 and 5, in which the population is selected for higher risk patients, were not pre-specified analyses.

Minor comments

- In introduction, the authors do not discuss the significance of determining the the association between BRCA family history and increased aggressiveness. These patients should undergo germline testing themselves as the next step in management, so knowing family history doesn't seem clinically relevant alone. This is partially addressed in the Discussion.

- Introduction, line 41-42. Need a reference for "PC is a heterogeneous malignancy."

- Methods, is Gleason score based on biopsy or self-report

- Methods, "presence of PSA at diagnosis" is unclear. Do the authors mean presence of elevated PSA? Or the median?

- Methods, what is a "high tumor stage?" T2 or above? T3 or above? T4?

- Methods, it is unclear why tumor images were "redefined before images," or why tumor images are being looked at at all.

- Methods, clearly define dependent vars, independent vars, and covariates.

- Methods, how missing data will be handled is not discussed.

- Methods line 128, "high risk" is a better term than "highly potent"

- A number of language errors are noted throughout, e.g. "Additional clinical data were extracted from the electronic review chart." Copyeditor review is needed.

- An example of the FCH questionnaire should be included as Supplementary information.

Reviewer #2: My overarching concern with this manuscript is that the authors propose to look at the relationship between BRCA1/2-related family history and high Gleason score, yet these two are already linked because NCCN guidelines for genetic counseling referral include "high risk" cancers that are defined in part by high Gleason scores. There are also several methodological issues that are concerning:

1. How is BRCA-related family history defined? First degree relatives, second, third, etc?

2. If this is a cross-sectinoal study, there cannot be loss to follow-up.

3. Were questionnaires administered at the time of diagnosis or after? This has substantial implications for survivor or selection bias, particularly for cases diagnosed in the earlier timeframe.

4. What was the distribution of year of diagnosis for the 473 patients who responded with sufficient data compared to the rest of the cohort?

5. Table 1 should include a column for the remaining individuals in the cohort to determine whether there are substantial differences in personal or tumor characteristics among those included in the study (i.e. selection bias)

6. Why not consider the NCCN definition of a suggestive family history (father or brother or multiple relatives with prostate cancer diagnosed under age 60, at least one relative with breast, ovarian, or pancreatic cancer, at least one relative with colorectal, ovarian, pancreatic, or kidney cancer)?

6. PLOS authors have the option to publish the peer review history of their article (what does this mean?). If published, this will include your full peer review and any attached files.

Reviewer #1: No

Reviewer #2: No

---

## [Author Response · Author response to Decision Letter 0]

22 Sep 2020

We appreciate the effort and constructive feedback provided regarding our manuscript, and hope that our edits and the corresponding responses address all of the issues and concerns that have been raised and that our manuscript will now be considered suitable for publication in PLOS ONE.

Academic Editor

1. Potential for bias in ascertainment of participants (unusually high overall Gleason scores) as well as in follow-up. It is of critical importance to address specific concerns about study bias raised by the reviewers.

Response: We recognize that this is a substantial concern for this study as the reviewers have commented. 

For the Gleason score, there is an overlap in patients with a Gleason score ≥ 8 (38.0%) and ≥ 9 (19.4%); thus, a total of 38% of our cohort had Gleason score ≥ 8. We believe this is within the expected range. We have revised the associated text in the manuscript (page #8, line #147-148). 

As for the questionnaires and the follow-up, the questionnaires were administered to each patient on their follow-up clinic visits during May to September 2017. As our follow-up protocol for prostate cancer consists of a visit every three months minimum, we assumed that all patients currently followed at our department were theoretically provided with the questionnaire (page #4, lines #83-#88).

Furthermore, we agree that patient survival and selection bias is a major concern in this protocol. We have added statements regarding this as a limitation in the revised Discussion (page #18, lines #304-#310).

2. The definition of "high-risk" being based entirely on Gleason score. See reviewer's comments on other factors to include.

Response: As you and the reviewer #1 have pointed out, we agree that multiple factors are applied for the classification of aggressiveness in clinically localized cancers. However, we managed to gather data from all prostate cancer patients as long as they were willing to provide their family data. Consequently, those with metastatic disease were also included. As PSA and T stage are generally affected by timing of diagnosis and concomitant tumor burden in metastatic disease, we set Gleason grade as a single primary surrogate for aggressiveness. 

The number of patients with metastatic disease at initial visit was extremely small (n = 9), which we presume is due to the structure of our clinic in that most of the patients were referred to us from affiliated health-checkup centers following detection of elevated PSA. We analyzed whether the presence of metastatic disease is associated with BRCA family cancer history and found no significant relationship between two. The results are described in the Results (page #10, lines #181-#184) as well as Table 3. 

We also conducted an additional analysis for the cohort with localized prostate cancer (PC). In this analysis, we assessed the relationship between BRCA FCH and the High-Risk Group based on NCCN classification, high (≥ 8) Gleason score, and PSA at diagnosis. The results for which have been added to the revised Results section (page #11, line #114- page #12, line #205). Briefly, the results were similar to that of the entire population. Patients with BRCA-associated FCH were less likely to present with high cT stage, and had higher BMI (though this seemed invaluable clinically). No direct association between BRCA-associated FCH and GS ≥ 8, or NCCN high risk classification, was observed via univariate or multivariate analysis. 

We have also added relevant discussion regarding this analysis in the revised manuscript (page #15, lines #237-#240). 

3. There are already clinical recommendations in the United States that suggest that men with high risk prostate tumors undergo genetic testing for BRCA2 as they are tightly correlated. In the introduction and background these studies, as well as the NCCN guidelines, should be references and results here should be put into the context of these data.

Response: Thank you for your constructive suggestion. 

We see that there are already clinical recommendations in the U.S. as NCCN guidelines recommend including high Gleason score in criteria for genetic counseling referral. We agree that these “high-risk” patients should ultimately undergo germline testing. However, as we have discussed in the Discussion section, it is challenging to perform genetic aberration analysis in Japan due to both the high associated costs and the issues surrounding genetic counseling. Our understanding is that it remains unclear if family cancer history itself adds information aside from persuading genetic testing. Hence, if any relationship between family cancer history and aggressiveness is discovered, it would benefit Japanese patients and clinicians. 

We have revised the Introduction according to your comments to adequately incorporate both your idea and our objectives (page #3, line #65 - page #4, line #72). We hope the revised Introduction better serves to highlight the background and the purpose of our study. 

4. BRCA1 and BRCA2 appear to be considered similarly. Existing evidence suggests that BRCA2 is much more likely linked to aggressive prostate cancer (and prostate cancer in general) compared to BRCA1. Re-assess your data in this context and update your background and discussion to better highlight these differences.

Response: We agree that distinction between BRCA1 and BRCA2 is important. We also agree that clinically, differentiation between the two is relatively difficult. In fact, clinical trials evaluating the PARP inhibitor, Olaparib, for metastatic prostate cancer, used both BRCA1 and BRCA2 (and ATM also) as inclusion criteria[1, 2]. Therefore, we believe treating BRCA1 and BRCA2 as a group when looking at their family history is a valid clinical practice. As per your suggestion, we have added text addressing the distinction between the two genes in the revised manuscript (page #3, lines #50-#53 and page #16, line #265 - page #17, line #271).

5. More details on how families were selected for inclusion and details on the factors to be considered in the multivariate model.

Response: 

In this study, prostate, breast, ovarian and pancreatic cancers were considered BRCA-associated cancers, and FCH within two-degree relatives were collected. 

We selected variables that had been previously reported to influence prostate cancer severity or survival. Systemic review has suggested that smoking partially impacts patient survival. We have incorporated your comment regarding the variables in the multivariate model, as well as how the following variables were used for each analysis.

-Analysis of association between GS and variates (Table 2): age, BMI, smoking, PSA, high T stage, BRCA FCH, Other FCH.

-Analysis of association between BRCA FCH and variates (Table 3): age, BMI, smoking, PSA, high T stage, GS ≥ 8, Other FCH. We removed GS ≥ 9 from the analysis to improve the clarity of the results. 

The Patients and methods (page #5, lines #103-#11) section, as well as the entire Results section has been revised accordingly. 

6. Address the other points raised by the reviewers.

Response: We have worked to incorporate the comments provided by you and the two reviewers into our revised manuscript. Again, your feedback is highly appreciated and we hope these revisions persuade you to accept our submission for publication.

 

Reviewer #1: This is a retrospective cross sectional study of Japanese men with prostate cancer, and whether presence of BRCA-associated family history is associated with aggressiveness of prostate cancer based on PSA, Gleason Grade, and T stage at diagnosis. They did not find any association between BRCA fam history and cancer aggressiveness. The topic is novel & timely, and the manuscript is overall well written aside from minor language errors. However, several methodological flaws limit validity of findings.

Response: We appreciate the effort and constructive feedback provided regarding our manuscript.

Major comments

- The authors refer to "BRCA genes," but BRCA is a family of genes that includes BRCA1 and BRCA2. The distinction is important as BRCA1 variant carriers have only a slightly greater risk of prostate cancer than non-carriers, versus BRCA2 where the risk of prostate cancer is 7-fold. This may have implications on prostate cancer aggressiveness. Though it is difficult to differentiate a BRCA1-family history vs BRCA2-family history, the authors should nevertheless address this.

Response: We agree that distinction between BRCA1 and BRCA2 is important. We also agree that clinically, differentiation between the two is relatively difficult. In fact, clinical trials evaluating the PARP inhibitor, Olaparib, for metastatic prostate cancer, used both BRCA1 and BRCA2 (and ATM also) as inclusion criteria[1, 2]. Therefore, we believe treating BRCA1 and BRCA2 as a group when looking at their family history is a valid clinical practice. As per your suggestion, we have added text addressing the distinction between the two genes in the revised manuscript (page #3, lines #50-#53 and page #16, line #265 - page #17, line #271).

- It is difficult to grasp the main message in the Discussion.

Response: Our original outcomes of interest were prostate cancer severity, and defined Gleason score ≥ 8, which we failed to show any significant association for. Our other primary objective of this study was to clarify BRCA-associated family cancer history in a Japanese cohort, as this have not been previously reported to date, and to validate whether our modified criteria for a highly potent cohort are valuable. Also included in the discussion is the potential issue concerning genetic counseling and testing in Japan, which prompted us to perform this study. 

We have edited the order of paragraphs in which each idea is stated to make the discussion clearer (page #15, line #241 to page #18, line #300). 

- The cohort-defining variable of "BRCA-associated FCH" is not defined in the Methods. Specifically, which cancers and confirming Carter et al wa sused.

Response: In this study, prostate, breast, ovarian and pancreatic cancers were considered BRCA-associated cancers, and FCH within two-degree relative were collected. We used the definition described by Carter et al as a reference. 

We have added this information to the revised Methods (page #4, line #88 - page #5, line #92)

- The primary endpoint of Gleason score is not justified. In the USA, a composite "aggressiveness" score including Gleason, PSA, and T stage are combined to create a Grade Group which is used to make management decisions - never one alone. Similarly, it is unclear why regional and metastatic disease are not included as "aggressive" prostate cancer.

Response: Thank you for providing these insights. We agree that multiple factors are applied for the classification of aggressiveness in clinically localized cancers. However, we managed to gather data from all prostate cancer patients as long as they were willing to provide their family data. Consequently, those with metastatic disease were also included. As PSA and T stage are generally affected by timing of diagnosis and concomitant tumor burden in metastatic disease, we set Gleason grade as a single primary surrogate for aggressiveness. 

The number of patients with metastatic disease at initial visit was extremely small (n = 9), which we presume is due to the structure of our clinic in that most of the patients were referred to us from affiliated health-checkup centers following detection of elevated PSA. We analyzed whether the presence of metastatic disease is associated with BRCA family cancer history and found no significant relationship between two. The results are described in the Results (page #10, lines #181-#184) as well as Table 3. 

We also conducted an additional analysis for the cohort with localized prostate cancer (PC). In this analysis, we assessed the relationship between BRCA FCH and the High-Risk Group based on NCCN classification, high (≥ 8) Gleason score, and PSA at diagnosis. The results for which have been added to the revised Results section (page #7, lines #2-#10). Briefly, the results were similar to that of the entire population. Patients with BRCA-associated FCH were less likely to present with high cT stage, and had higher BMI (though this seemed invaluable clinically). No direct association between BRCA-associated FCH and GS ≥ 8, or NCCN high risk classification, was observed via univariate or multivariate analysis. 

We have also added relevant discussion regarding this analysis in the revised manuscript (page #15, lines #237-#240). 

- There is high potential for selection bias. It is not described in the methods to whom/how/when the questionnaire that ultimately defines family history is sent. Only 20% of patients were eligible and evaluable for the outcome, but this is not quantified in Results. On my review, it does appear that there is bias: 57% of patients had a Gleason score of 8 or higher, which is incredibly high. The finding of association of BRCA Fam History with *early* tumor stage is inconsistent with prior studies of positive known BRCA variant carriers as the authors mention, which also raises the question of bias. The authors admit that it is challenging to estimate aggressive based on family history information only. Moreover, the lack of BRCA mutation status in the study cohort is a major limitation.

Response: We recognize this is a substantial concern for this study. 

The questionnaires were administered to each patient on their follow-up clinic visits during May to September 2017. As our follow-up protocol for prostate cancer consists of a visit every three months minimum, we assumed that all patients currently followed at our department were theoretically provided with the questionnaire (page #4, lines #83-#88).

We also agree that the relatively low percentage of patients who returned the questionnaire is a major concern and serves as a limitation in this study. We have clarified the percentage in the Results (page #6, lines #133-#134).

We decided that it was necessary to reconfirm the family history for each patient, partly because pancreas cancer history was not documented in daily practice. Therefore, it was prioritized to ensure that patients included in the analysis could be reached at the time of analysis and have reliable memory. As we extracted all prostate cancer cases during the initial screening process the proportion of included patients was relatively low. However, we believe most of those who fulfilled the above-mentioned criteria were prioritized and included in the analyzed cohort. 

There is an overlap in patients with a Gleason score ≥ 8 (38.0%) and ≥ 9 (19.4%); thus, a total of 38% of our cohort had Gleason score ≥ 8. We believe this is within the expected range. This section has been revised in the manuscript (page #8, line #147-148).

Your assessment that a relationship between BRCA family history and early tumor stage is not in line with previous studies, is valid. As is noted in the Discussion, we believe that the most reasonable answer for this finding is that patients with a family history have a greater interest in screening to have their cancer detected earlier. We partly agree that there is a potential for selection bias; however, the idea mentioned, as well as the difficulty associated with estimating severity or aggressiveness based on family history information alone, in itself would be of help when planning future studies regarding family history of malignancy, which we have addressed in the revised Discussion section (page #16, lines #254-#257). 

We also agree that lack of BRCA gene testing is a major limitation, which we have noted in the Discussion, genetic testing is currently extremely challenging in Japan which causes fewer people to undergo counseling in the first place. This is the reason why we conducted additional analysis to identify highly potent candidates. Although there is a need for advanced gene testing in Japan (which is beyond this study), we believe that the ability to narrow down the potential candidates for detection via genetic testing is of considerable interest to clinicians in Japan. We have, therefore, retained this argument in the Discussion. 

- It is not described how the variables in the multivariate model are selected. Table 2 includes 11 variables, some of which are co-linear (e.g. BRCA FCH and Prostate) which does not seem appropriate. Why is smoking status included in the model?

Response: 

We selected variables that had been previously reported to influence prostate cancer severity or survival. Systemic review has suggested that smoking partially impacts patient survival. We have incorporated your comment regarding the variables in the multivariate model, as well as how the following variables were used for each analysis (page #4, lines #16-#23).

-Analysis of association between GS and variants (Table 2): age, BMI, smoking, PSA, high T stage, BRCA FCH, Other FCH.

-Analysis of association between BRCA FCH and variates (Table 3): age, BMI, smoking, PSA, high T stage and GS ≥ 8 and metastasis at diagnosis. 

-Analysis of association between BRCA FCH and variates for localized cancer (added; Table 4): age, BMI, smoking, high NCCN risk, PSA, high T stage and GS ≥ 8.

We removed GS ≥ 9 from the analysis to improve the clarity of the results. The Results are revised and rewritten (page #8, line #159 to page #12, line #205). 

- Several analyses are not described in the methods. For example, Table 3 is reported but unclear what this adds to the message. Tables 4 and 5, in which the population is selected for higher risk patients, were not pre-specified analyses.

Response: We have revised the Patients and methods section and have included a new subsection “Study protocol and outcomes”. Here, we have described that relationships between BRCA family cancer history and other clinically considered variables were analyzed (page #5, lines #101-#111). We also added methods on how we attempted to identify highly potent candidates for BRCA in this section (page #6, lines #115-#122). 

Minor comments

- In introduction, the authors do not discuss the significance of determining the the association between BRCA family history and increased aggressiveness. These patients should undergo germline testing themselves as the next step in management, so knowing family history doesn't seem clinically relevant alone. This is partially addressed in the Discussion.

Response: As we have discussed in the Discussion section, it is incredibly challenging to perform genetic aberration analysis in Japan due to the high associated costs and issues surrounding genetic counseling. Hence, if any relationship were identified between family cancer history and aggressiveness it would prove beneficial to Japanese patients and clinicians. 

We agree with your viewpoint that these patients should ultimately undergo germline testing. Our suggestion when drafting this study protocol was that defining family history may be useful for both predicting aggressive cancer and defining highly potent cohorts for germline screening. 

We have added information regarding these points in the revised introduction (page #4, lines #67-#72).

- Introduction, line 41-42. Need a reference for "PC is a heterogeneous malignancy."

Response: A relevant reference has been added (page #3, lines #47-#48).

- Methods, is Gleason score based on biopsy or self-report

Response: The Gleason score is based on biopsy report, and all diagnoses were made by a single team of pathologists. We have added a sentence regarding this (page #5, lines #113 - #114).

- Methods, "presence of PSA at diagnosis" is unclear. Do the authors mean presence of elevated PSA? Or the median?

Response: We agree that this sentence was vague. We have, therefore, clarified that median PSA was the outcome (page #5, lines #104 and #108).

- Methods, what is a "high tumor stage?" T2 or above? T3 or above? T4?

Response: We agree that this sentence was vague. We have clarified that T3a or higher was defined as a “high tumor stage” (page #5, line #105).

- Methods, it is unclear why tumor images were "redefined before images," or why tumor images are being looked at all.

Response: We used clinical T stage as a dependent variable since pathological T stage was not available for all patients (lack of radical prostatectomy). Clinical T staging were based on digital rectal examination and MRI images as a supplement. 

For patients with available FCH, we retrospectively consulted both the electric medical charts and MRI images to determine if the staging was appropriate. In questionable cases a decision was made by two clinicians (radiologist and urologist). 

We have rewritten the Methods section to make this clearer (page #5, lines #111-#113).

- Methods, clearly define dependent vars, independent vars, and covariates.

Response: Dependent variables, independent variables and covariates are now defined in the Methods (page #5, lines #102-#111). 

- Methods, how missing data will be handled is not discussed.

Response: 

First, we have added a description regarding the patient selection method which included the original identification of 2, 286 patients as pathologically positive for prostate cancer between July 2003 and September 2017 by automatically screening electric medical charts. The following information was obtained for all 2, 286 patients (if available) for use in analysis: patient ID, age at diagnosis, date of diagnosis, body mass index, smoking history, subsequent radical prostatectomy or not, PSA at date nearest to cancer diagnosis and Gleason score. However, since clinical T staging could not be extracted from the medical chart database as it was in the form of narrative text and where to specify was not standardized throughout the screened period. Per our original study design, we requested permission for descriptive chart review for those with returned questionnaires only, and the Internal Ethics Review Board approved the study as such. Thus, clinical T stage for the remaining individuals was not available. 

We have rewritten the “Patients and data collection” section (page #4, lines #79-#88) and moved some of the text in the “Patient characteristics” section (page #6, lines #133- page #7, lines #140) to make this process clearer and easier to understand. 

To visualize information for the population with missing family cancer history data, available patient characteristics were added to the Results (page #6, lines #133- page #7, lines #140), and a column was added in Table 1 for the remaining clinical information.

- Methods line 128, "high risk" is a better term than "highly potent"

Response: We agree with your comment and the term has now been revised per your suggestion. We appreciate your feedback.

- A number of language errors are noted throughout, e.g. "Additional clinical data were extracted from the electronic review chart." Copyeditor review is needed.

Response: Thank you for this comment. The final version of the revised manuscript is now reviewed and grammer-checked by professional English editing service (http://www.editage.jp). We hope the quality now satisfies publication criteria for PLOS ONE. 

- An example of the FCH questionnaire should be included as Supplementary information.

Response: We appreciate your suggestion. The English translation of the original questionnaire is now added as a supporting file (S1 Fig). 

Reviewer #2: My overarching concern with this manuscript is that the authors propose to look at the relationship between BRCA1/2-related family history and high Gleason score, yet these two are already linked because NCCN guidelines for genetic counseling referral include "high risk" cancers that are defined in part by high Gleason scores. There are also several methodological issues that are concerning:

Response: First of all, we appreciate the effort and constructive feedback provided regarding our manuscript. 

We agree that the association between the presence of BRCA 1/2 mutations and prevalence or aggressiveness of prostate cancer has been previously reported, and that these studies looked at patients with confirmed BRCA 1/2 mutation at the time of enrollment. Indeed, it is based on the results of these studies that the NCCN guidelines recommendation now include high Gleason score as a criteria for genetic counseling referral. However, our understanding is that it remains unclear whether family cancer history itself provides additional information aside from persuading genetic testing. 

We agree that these “high risk” patients should ultimately undergo germline testing. However, as we have discussed in the Discussion section, it is very challenging to perform genetic aberration analysis in Japan due to the high associated costs and issues surrounding genetic counseling. Hence, if any relationship between family cancer history and aggressiveness is discovered, it would benefit Japanese patients and clinicians.

We have added statements regarding these ideas in the revised Introduction (page #3, line #65- page #4, line #72).

1. How is BRCA-related family history defined? First degree relatives, second, third, etc?

Response: In this study, prostate, breast, ovarian and pancreatic cancers were considered BRCA-associated cancers, and FCH within two-degree relatives were collected. We used the definition described by Carter et al as a reference. 

We have added this statement in the Methods (page #4, line #88 - page #5, line #92)

2. If this is a cross-sectinoal study, there cannot be loss to follow-up.

Response: We agree that the term “lost to follow” is inappropriate for cross sectional study. What we meant was that patients who were no longer followed at our hospital were excluded. We have rewritten this section (page #4, lines #82-#87) to make this clear.

3. Were questionnaires administered at the time of diagnosis or after? This has substantial implications for survivor or selection bias, particularly for cases diagnosed in the earlier timeframe.

Response: You have raised an important question. The questionnaires were administered to each patient on their follow-up clinical visits during May to September 2017. As our follow-up protocol for prostate cancer consists of a visit every three months minimum, we assumed that all patients currently followed at our department were theoretically provided with the questionnaire (page #4, lines #83-#88). We agree that patient survival or selection bias is a major concern in this protocol. We have revised statements regarding this limitation in the Discussion (page #18, lines #304-#310).

4. What was the distribution of year of diagnosis for the 473 patients who responded with sufficient data compared to the rest of the cohort?

Response: 

First, we have added a description regarding the patient selection method which included the original identification of 2, 286 patients as pathologically positive for prostate cancer between July 2003 and September 2017 by automatically screening electric medical charts. The following information was obtained for all 2, 286 patients (if available) for use in analysis: patient ID, age at diagnosis, date of diagnosis, body mass index, smoking history, subsequent radical prostatectomy or not, PSA at date nearest to cancer diagnosis and Gleason score. Of those, 301 had insufficient clinical data and were excluded. Thus, 1,985 remained as candidates for family cancel history querying. Those who returned the questionnaire were ultimately analyzed. We have rewritten the “Patients and methods” (page #4, lines #79-#88) to make this clear.

Among the 1,985 candidates, those in the responding cohort were diagnosed with PC later in the study period (median year, 2015) with more than 75% responding after 2010, compared with 2007 for the remaining cohort. Thus, patients who responded were diagnosed more recently and were actively followed by 2017. 

We presume this difference is purely due to our study protocol. We have incorporated this information in the Results (page #6, lines #134-#136) and the Discussion (page #18, lines #306-#307).

5. Table 1 should include a column for the remaining individuals in the cohort to determine whether there are substantial differences in personal or tumor characteristics among those included in the study (i.e. selection bias)

Response: Thank you for your suggestion. As is explained in #4, the following information was obtained for all 2, 286 patients (if available) for use in analysis: patient ID, age at diagnosis, date of diagnosis, body mass index, smoking history, subsequent radical prostatectomy or not, PSA at date nearest to cancer diagnosis and Gleason score. However, since clinical T staging could not be extracted from the medical chart database as it was in the form of narrative text and where to specify was not standardized throughout the screened period. Per our original study design, we requested permission for descriptive chart review for those with returned questionnaires only, and the Internal Ethics Review Board approved the study as such. Thus, clinical T stage for the remaining individuals was not available. A column was added in Table 1 for the remaining clinical information. 

6. Why not consider the NCCN definition of a suggestive family history (father or brother or multiple relatives with prostate cancer diagnosed under age 60, at least one relative with breast, ovarian, or pancreatic cancer, at least one relative with colorectal, ovarian, pancreatic, or kidney cancer)?

Response: 

As you have pointed out, NCCN guidelines suggest germline screening for prostate cancer patients with strong family history of prostate cancer and multiple (≥ 3) cancers on the same side of the family, particularly those diagnosed at ≤ 50 years of age (bile duct, breast, colorectal, endometrial, gastric, kidney, melanoma, ovarian, pancreatic, prostate (but not clinically localized Grade Group 1), small bowel, or urothelial cancer). This criteria targets mutation of both BRCA (1 and 2) and Lynch. In other words, some cancer types included are not cited for relationship with BRCA (1 and/or 2). Our study targets only family cancer history of BRCA 1/2 only, and this difference is the reason why we chose to define and utilize our own suggestion of BRCA family cancer history criteria. As our questionnaire was made to highlight these criteria, performing analysis by NCCN criteria as you suggest is impractical, and we believe that our original analysis would be more appropriate for this study’s vision.

We have added new sentences describing the reason we chose to define and utilize our own suggested criteria in the Introduction section (page #3, lines #60-#65).

---

## [Decision Letter · Decision Letter 1]

8 Oct 2020

PONE-D-20-24272R1

Association between prostate cancer characteristics and BRCA-associated family cancer history in a Japanese cohort

PLOS ONE

Dear Dr. Ishiyama,

Thank you for submitting your manuscript to PLOS ONE. After careful consideration, we feel that it has merit but does not fully meet PLOS ONE’s publication criteria as it currently stands. Therefore, we invite you to submit a revised version of the manuscript that addresses the points raised during the review process.

1.  In the methods define BRCA FCH.  See reviewer's comments.

2.  Include the 21% evaluable rate as a limitation in the abstract.

3.  Expand limitations in the discussion including the high proportion of patients with aggressive disease.  See reviewer's comments.

4.  Remove Tables 5 and 6 as these are based on small numbers and rationale for inclusion is not clear

5.  Make sure that all statements in the conclusion are fully supported by data.  See reviewer's comments.

6.  Address the other comments raised by the reviewers.

7.  The writing in the manuscript needs considerable editing.  See specific points raised by the reviewers.  Consider having additional people edit it for grammar and readability.

We look forward to receiving your revised manuscript.

Kind regards,

Amanda Ewart Toland, Ph.D.

Academic Editor

PLOS ONE

Reviewers' comments:

Reviewer's Responses to Questions

**Comments to the Author**

1. If the authors have adequately addressed your comments raised in a previous round of review and you feel that this manuscript is now acceptable for publication, you may indicate that here to bypass the “Comments to the Author” section, enter your conflict of interest statement in the “Confidential to Editor” section, and submit your "Accept" recommendation.

Reviewer #1: (No Response)

Reviewer #2: (No Response)

2. Is the manuscript technically sound, and do the data support the conclusions?

Reviewer #1: Partly

Reviewer #2: Yes

3. Has the statistical analysis been performed appropriately and rigorously? 

Reviewer #1: Yes

Reviewer #2: Yes

4. Have the authors made all data underlying the findings in their manuscript fully available?

Reviewer #1: Yes

Reviewer #2: Yes

5. Is the manuscript presented in an intelligible fashion and written in standard English?

Reviewer #1: No

Reviewer #2: Yes

6. Review Comments to the Author

Reviewer #1: The authors have made significant effort addressing the comments. However, not all comments are completely addressed, and some of the conclusions are not supported by the data. Lastly, the copyediting needs work; there are errors that make the paper cumbersome to read, and muddy some important points.

1. The 21% evaluable rate is appropriately mentioned in the discussion. However, it should also be mentioned in the abstract results since it's an important limitation. Also it should be noted in the discussion that the analyzed cohort is concentrated with patients with aggressive disease: 38% had Gleason 8 or greater. So the analyzed cohort may not be representative of the Japanese male prostate cancer population. For example, in this paper of Japanese men who underwent radical prostatectomy, only 16% had Gleason 8+ (PMID 19716590)

2. In Methods, please define positive BRCA FCH. This is important as the term is used in Tables 2-4. For example, does "positive" mean any single second degree family member with BRCA associated cancer? Or meeting your modified Carter criteria? If the latter, do you have to meet all 3 criteria? 2? 1?. Also, for the modified Carter criteria, do they have to be in the same side of the family? For criterion 3) are these first degree relatives? second?

3. In Results, the rationale behind Tables 5 and 6 is not strong since it is unclear why B/C (vs another grouping) was selected, especially when there are only 6 patients.

4. In Discussion, I agree with the authors' hypothesis that the reason there is an association with BRCA FCH and low T stage might be patient-driven earlier screening (as seen in the numerically lower rate of NCCN high risk stage).

5. In the conclusion, I suggest that "The incidence of PaC FCH was higher among the patients with PC than publicly anticipated" is not appropriate to include since it is not discussed in the Discussion and is likely due to selection bias. I would either remove the sentence or add a hedging phrase. Also, I would not mention that BRCA FCH may be a useful tool for cancer screening, as this study was not designed to evaluate this.

6. The writing is still not up to par. Examples of writing errors or awkward wording:

- Abstract first sentence: "besides" is used incorrectly. Either replace with "as well as" or "in addition to," or remove reference to breast/ovarian/pancreatic cancers altogether. I recommend reading https://wp.nyu.edu/sciwriabudhabi/2018/07/31/also-furthermore-moreover-besides/

- Abstract third sentence: "characterize" may be more appropriate than "clarify."

- Introduction: did Carter define hereditary risk as needing all three of those criteria? THis is what you suggest by saying "and" not "or."

- Line 51, "Meanwhile" is awkward.

- Line 53, Olaparib should not be capitalized.

- Line 65-66, unnecessary capitalizaton

etc.

Minor points:

Abstract last sentence: "This is the first report of BRCA-associated FCH" isn't true (it's the first report of its association with prostate cancer aggressiveness). Either remove or specify in this way or add "in Japanese men."

Introduction

- Line 61: NCCN recommends germline testing, not just genetic counseling, in this group.

Discussion:

- Lines paragraph 278-284 are speculative, as confirmation of BRCA1/2 carriage was not done.

- Reference to PARP inhibition is moot if germline testing cannot be done. Somatic testing would be the way to go. Also PARP inhibitors are used in mCRPC, which is outside the scope of this study, so not too relevant.

Reviewer #2: The authors' revisions have substantially improved this manuscript. I have a few remaining comments:

1. “BRCA gene” should be BRCA1/2 genes. Referring to both genes as just "BRCA" throughout the text is problematic.

2. Including the NCCN guidelines for genetic counseling in the introduction makes the argument less clear (lines 60-67). Maybe include this in methods for rationalizing definition of FCH used in this study. The last two sentences in lines 67-72 do help to clarify the goal of the manuscript.

3. From the manuscript alone, it is still unclear how you are defining BRCA-FCH. There should be a sentence to the effect of “BRCA-associated FCH was defined as at least one(?) first or second degree relative with breast, prostate, ovarian, or pancreatic cancer”.

4. High PSA at diagnosis and high clinical tumor stage are listed as covariates in lines 104-105, and then as outcomes in lines 108-109. Were these actually included as covariates?

5. P-values for table 1 would be helpful

6. Why are p-values missing in Table 2 for the individual cancer types?

7. Double-check p-values for BMI in Table 3. Those medians look very similar and the sample size isn’t large enough to detect such a small difference.

8. Table 5. Why only B/C vs rest? Why not include A? I’m not sure what Tables 5 and 6 add to the results.

7. PLOS authors have the option to publish the peer review history of their article (what does this mean?). If published, this will include your full peer review and any attached files.

Reviewer #1: No

Reviewer #2: No

---

## [Author Response · Author response to Decision Letter 1]

8 Nov 2020

Academic editor

1. In the methods define BRCA FCH. See reviewer's comments.

Response: Thank you for your comment. In this study, positive BRCA1/2-associated FCH was defined as a single second-degree family member with BRCA1/2-associated cancer (i.e. prostate, breast, pancreas, ovarian). We have clarified this definition by adding the following sentence in the methods: 

“PC, BC, OvC, and PaC were considered BRCA1/2-associated cancers. Positive BRCA1/2-associated FCH was defined as a single second-degree family member with such cancers, as per the recommendations of the National Comprehensive Cancer Network (NCCN) suggesting genetic testing for PC patients with a strong family history of PC and multiple (≥3) cancers of a particular type, and NCCN guidelines for genetic/familial high-risk assessment of PaC, BC, and OvC” (page #4, lines #81-#86).

2. Include the 21% evaluable rate as a limitation in the abstract.

Response: We agree that evaluable rate in this study is a major limitation, and have included this number in the abstract:

“Among the 1,985 eligible candidates, 473 (23.83%) patients had adequately detailed FCH, obtained via questionnaire, and were thus included in the study.” (page #2, lines #31-#33). 

3. Expand limitations in the discussion including the high proportion of patients with aggressive disease. See reviewer's comments.

Response: We agree that our cohort may not be an exact representation of the general Japanese population, as it is a single institution study. We have included the following sentences in the discussion: 

“Indeed, 38% of patients had GS ≥ 8, higher than previously reported for Japanese PC patients [26]. Thus, to what extent our cohort precisely represents the current Japanese PC population remains unclear.” (page #17, lines #304-#306). 

The study mentioned by reviewer #1 was added in the reference section.

4. Remove Tables 5 and 6 as these are based on small numbers and rationale for inclusion is not clear

Response: We appreciate your feedback. According to your suggestion, we have decided to delete Tables 5 and 6.

5. Make sure that all statements in the conclusion are fully supported by data. See reviewer's comments.

Response: Thank you for pointing this out. According to suggestions made by you and the reviewers, changes have been made so the conclusions are fully supported by our data. The conclusion has been revised to the following: 

“In conclusion, this is the first descriptive report of complete BRCA1/2-associated FCH among patients with PC in Japan. BRCA1/2-associated FCH was not associated with severity or aggressiveness of PC. Further investigation of this population and FCH of related cancers may uncover candidates with high potential for BRCA1/2 mutations.” (page #18, lines #320-#323).

6. Address the other comments raised by the reviewers.

Response: We have worked to incorporate the suggestions provided by you and the two reviewers into our revised manuscript R2. Your feedback is highly appreciated, and we hope these revisions persuade you to accept our submission for publication.

7. The writing in the manuscript needs considerable editing. See specific points raised by the reviewers. Consider having additional people edit it for grammar and readability.

Response: We appreciate your advice. The revised manuscript (R2) has gone through double-copyediting by an English-native U.S. certified physician and a professional English editing service. We hope the new version of our manuscript now meets the standards for publication in PLOS ONE. 

 

Reviewer #1: The authors have made significant effort addressing the comments. However, not all comments are completely addressed, and some of the conclusions are not supported by the data. Lastly, the copyediting needs work; there are errors that make the paper cumbersome to read, and muddy some important points.

Response: We appreciate your thoughtful feedback regarding our R1 manuscript. Issues pointed out by you and other reviewers have now been addressed in the main text and changes have also been made so the conclusions are fully supported by our data. 

The re-revised manuscript (R2) has gone through double-copyediting by an English-native U.S. certified physician and a professional English editing service. 

We hope the new version of our manuscript now meets the standards for publication in PLOS ONE. 

1. The 21% evaluable rate is appropriately mentioned in the discussion. However, it should also be mentioned in the abstract results since it's an important limitation. Also it should be noted in the discussion that the analyzed cohort is concentrated with patients with aggressive disease: 38% had Gleason 8 or greater. So the analyzed cohort may not be representative of the Japanese male prostate cancer population. For example, in this paper of Japanese men who underwent radical prostatectomy, only 16% had Gleason 8+ (PMID 19716590)

Response: Thank you for this feedback. 

We agree that the evaluable rate in this study is a major limitation, and have included this point in the abstract: 

“Among the 1,985 eligible candidates, 473 (23.83%) patients had adequately detailed FCH, obtained via questionnaire, and were thus included in the study.” (page #2, lines #31-#33). 

We also agree that our cohort may not be an exact representation of the general Japanese population, as it is a single institution study. We have included the following sentence in the discussion: 

“Indeed, 38% of patients had GS ≥ 8, higher than previously reported for Japanese PC patients [26]. Thus, to what extent our cohort precisely represents the current Japanese PC population remains unclear.” (page #17, lines #304-#306). 

The study you mentioned was added in the reference section.

2. In Methods, please define positive BRCA FCH. This is important as the term is used in Tables 2-4. For example, does "positive" mean any single second degree family member with BRCA associated cancer? Or meeting your modified Carter criteria? If the latter, do you have to meet all 3 criteria? 2? 1?. Also, for the modified Carter criteria, do they have to be in the same side of the family? For criterion 3) are these first degree relatives? second?

Response: Thank you for your comment. In this study, positive BRCA1/2-associated FCH was defined as a single second-degree family member with BRCA1/2-associated cancer (i.e. prostate, breast, pancreas, ovarian). We have clarified this definition by adding the following sentence in the methods: 

“PC, BC, OvC, and PaC were considered BRCA1/2-associated cancers. Positive BRCA1/2-associated FCH was defined as a single second-degree family member with such cancers, as per the recommendations of the National Comprehensive Cancer Network (NCCN) suggesting genetic testing for PC patients with a strong family history of PC and multiple (≥3) cancers of a particular type, and NCCN guidelines for genetic/familial high-risk assessment of PaC, BC, and OvC” (page #4, lines #81-#86).

For criterion 3) in the modified Carter criteria, “two relatives < 55 years of age, within the second degree, diagnosed with BRCA-associated cancer” were considered a positive BRCA1/2-associated FCH. We have revised the sentence accordingly (page #6, lines #117-#118).

3. In Results, the rationale behind Tables 5 and 6 is not strong since it is unclear why B/C (vs another grouping) was selected, especially when there are only 6 patients.

Response: We appreciate your feedback, and agree that the rationale is weak. According to your comment and the editor’s suggestion, we have decided to delete Tables 5 and 6.

4. In Discussion, I agree with the authors' hypothesis that the reason there is an association with BRCA FCH and low T stage might be patient-driven earlier screening (as seen in the numerically lower rate of NCCN high risk stage).

Response: Thank you for this comment. We believe it is meaningful to show the behavioral pattern of cancer screening among Japanese males.

5. In the conclusion, I suggest that "The incidence of PaC FCH was higher among the patients with PC than publicly anticipated" is not appropriate to include since it is not discussed in the Discussion and is likely due to selection bias. I would either remove the sentence or add a hedging phrase. Also, I would not mention that BRCA FCH may be a useful tool for cancer screening, as this study was not designed to evaluate this.

Response: Thank you for your suggestion. We agree that referring to PaC FCH prevalence in the conclusion is not appropriate, and the sentence has now been removed. In the discussion, a brief statement regarding the relatively high percentage of PaC FCH was added: 

“Interestingly, 6.6% of our study population had an FCH of PaC, which was larger than in previous reports.” (page #16, lines #261-#262). 

According to your comment that referring to BRCA FCH as a useful tool for cancer screening is beyond the scope of this study, the relevant sentence has also been removed.

The conclusion has been revised to the following: 

“In conclusion, this is the first descriptive report of complete BRCA1/2-associated FCH among patients with PC in Japan. BRCA1/2-associated FCH was not associated with severity or aggressiveness of PC. Further investigation of this population and FCH of related cancers may uncover candidates with high potential for BRCA1/2 mutations.” (page #18, lines #320-#323).

6. The writing is still not up to par. Examples of writing errors or awkward wording:

- Abstract first sentence: "besides" is used incorrectly. Either replace with "as well as" or "in addition to," or remove reference to breast/ovarian/pancreatic cancers altogether. I recommend reading https://wp.nyu.edu/sciwriabudhabi/2018/07/31/also-furthermore-moreover-besides/

- Abstract third sentence: "characterize" may be more appropriate than "clarify."

- Introduction: did Carter define hereditary risk as needing all three of those criteria? THis is what you suggest by saying "and" not "or."

- Line 51, "Meanwhile" is awkward.

- Line 53, Olaparib should not be capitalized.

- Line 65-66, unnecessary capitalizaton

etc.

Response: Thank you for providing these insights. We have incorporated your suggested corrections into our main text. For hereditary PCa criteria by Carter et al, correct phrasing would be “or” instead of “and” as you have pointed out, and the sentence has been revised to the following:

“Accordingly, hereditary risk for PC has been defined as 1) more than three first-degree relatives with PC, 2) three successive generations with PC, or 3) two relatives affected under 55 years of age” (page #3, lines #46-#48).

To meet the writing standard for publication in PLOS ONE, the re-revised manuscript (R2) has gone through double-copyediting by an English-native U.S. certified physician and a professional English editing service. We hope you now find the manuscript grammatically solid. 

Minor points:

Abstract last sentence: "This is the first report of BRCA-associated FCH" isn't true (it's the first report of its association with prostate cancer aggressiveness). Either remove or specify in this way or add "in Japanese men."

Response: Thank you for your specific suggestion. According to your comment, the phrase "in Japanese men" has been added to the relevant sentence (page #2, lines #40-#42).

Introduction

- Line 61: NCCN recommends germline testing, not just genetic counseling, in this group.

Response: Thank you for pointing this out. This sentence is now revised in the methods as the following: 

“Positive BRCA1/2-associated FCH was defined as a single second-degree family member with such cancers, as per the recommendations of the National Comprehensive Cancer Network (NCCN) suggesting genetic testing for PC patients with a strong family history of PC and multiple (≥3) cancers of a particular type, and NCCN guidelines for genetic/familial high-risk assessment of PaC, BC, and OvC” (page #4, lines #82-#86).

Discussion:

- Lines paragraph 278-284 are speculative, as confirmation of BRCA1/2 carriage was not done.

Response: You have raised an important point. We agree that this part of the discussion is partly speculative as no genetic testing was carried out. However, we believe it is reasonable to estimate the percentage of genetic mutation holders from the results of previous reports. With no genetic testing performed, the only result we are able to present is the percentage of patients included in our novel risk model (A, B or C). As such, we would like to propose our risk model as a candidate for future use, but the actual validity of the model warrants further study. 

The relevant sentences in the discussion have been revised as the following: 

“These results implied that roughly 1-4% of all PC patients possess BRCA1/2 mutations, which is substantially lower than the 5.50% of patients included in one of our novel risk factor groups (A, B, or C). However, as we were unable to conduct confirmatory genetic testing in this study, the accuracy of our risk factor model warrants further validation.” (page #16, lines #278-#281). 

-Reference to PARP inhibition is moot if germline testing cannot be done. Somatic testing would be the way to go. Also PARP inhibitors are used in mCRPC, which is outside the scope of this study, so not too relevant.

Response: We agree that referring to PARP inhibitors as a possible benefit of FCH in the discussion is not justified. The relevant part has been replaced by the following sentence referring to the importance of FCH as a first step toward somatic germline testing: 

“Obtaining concrete FCH data offers several possible benefits. Most obviously, it is the initial step required to obtain somatic genetic testing, recommended for at-risk patients” (page #16, lines #282-#284).

 

Reviewer #2: The authors' revisions have substantially improved this manuscript. I have a few remaining comments:

Response: We appreciate your thoughtful feedback regarding our R1 manuscript. 

1. “BRCA gene” should be BRCA1/2 genes. Referring to both genes as just "BRCA" throughout the text is problematic.

Response: Thank you for this suggestion. We agree with the proposal and the term “BRCA gene” has been corrected to “BRCA1/2 genes” throughout the manuscript. 

2. Including the NCCN guidelines for genetic counseling in the introduction makes the argument less clear (lines 60-67). Maybe include this in methods for rationalizing definition of FCH used in this study. The last two sentences in lines 67-72 do help to clarify the goal of the manuscript.

Response: Thank you for this advice. We have moved and edited the mentioned part of the introduction into the methods section to rationalize the definition of BRCA1/2-associated FCH in this study. This new version of the sentence is as follows: 

“Positive BRCA1/2-associated FCH was defined as a single second-degree family member with such cancers, as per the recommendations of the National Comprehensive Cancer Network (NCCN) suggesting genetic testing for PC patients with a strong family history of PC and multiple (≥3) cancers of a particular type, and NCCN guidelines for genetic/familial high-risk assessment of PaC, BC, and OvC.” (page #4, lines #82-#86).

3. From the manuscript alone, it is still unclear how you are defining BRCA-FCH. There should be a sentence to the effect of “BRCA-associated FCH was defined as at least one(?) first or second degree relative with breast, prostate, ovarian, or pancreatic cancer”.

Response: Thank you for your comment. In this study, positive BRCA1/2-associated FCH was defined as “a single second-degree family member with BRCA1/2-associated cancer (i.e. prostate, breast, pancreas, ovarian). We have clarified this definition by adding the following sentence in the methods: 

“PC, BC, OvC, and PaC were considered BRCA1/2-associated cancers. Positive BRCA1/2-associated FCH was defined as a single second-degree family member with such cancers, as per the recommendations of the National Comprehensive Cancer Network (NCCN) suggesting genetic testing for PC patients with a strong family history of PC and multiple (≥3) cancers of a particular type, and NCCN guidelines for genetic/familial high-risk assessment of PaC, BC, and OvC” (page #4, lines #81-#86).

4. High PSA at diagnosis and high clinical tumor stage are listed as covariates in lines 104-105, and then as outcomes in lines 108-109. Were these actually included as covariates?

Response: Thank you for your question. 

In the analysis of factors contributing to high (≥ 8) GS, we included PSA and clinical T stage as covariates. This was because in this analysis, the dependent variable was high GS, the independent variable was FCH (all BRCA1/2-associated cancers, including prostate, breast, pancreatic, and ovarian cancers), and all other factors were defined as covariates. 

On the other hand, in the analysis of BRCA1/2-associated FCH and clinical factors, we defined the dependent variable as BRCA1/2-associated FCH and the independent variables as high GS, PSA, and clinical T stage. All other factors were defined as covariates. 

To clarify this idea, the relevant sentence has been revised to the following: 

“First, associations were determined between our independent variable, FCH (recorded as any BRCA1/2-associated cancer, as well as by PC, BrC, PaC, and OvC cancer type), and the primary dependent variable, high (≥ 8) Gleason score (GS). Covariates in this analysis included: age at diagnosis, body mass index (BMI), smoking history, prostate specific antigen (PSA) level at diagnosis, high clinical tumor (cT) stage defined as ≥ cT3a per NCCN Practice Guidelines in Oncology (Ver 2.2017), and the presence of other FCH not included in the dependent variable above. Next, patients were classified into two groups: those with and without BRCA1/2-associated FCH. We then compared the following dependent variables: high GS, PSA level at diagnosis (median), and high cT stage. Covariates included: age at diagnosis, BMI, smoking history, and presence of other FCH based on previously reported factors shown to influence PC severity or survival.” (page #5, lines #98-#108). 

5. P-values for table 1 would be helpful

Response: Thank you for your suggestion. P-values were added for the following factors: age at diagnosis, BMI, history of smoking, radical prostatectomy, PSA at diagnosis, GS ≥ 8, and GS ≥ 9. Information on clinical stage and presence of metastasis was not available for the “Remaining patients” cohort. 

6. Why are p-values missing in Table 2 for the individual cancer types?

Response: Thank you for pointing this out. P-values for univariate analysis for each cancer type have been added to Table 2 according to your suggestion. P-values for multivariate analysis are shown in Tables S1-S4 as these were calculated using four different multivariable models. 

7. Double-check p-values for BMI in Table 3. Those medians look very similar and the sample size isn’t large enough to detect such a small difference.

Response: You have raised an important point. We also questioned this result and repeated the analysis multiple times. However, this difference existed statistically, although it is apparently clinically meaningless. Therefore, the relevant result has been reported in Table 3. 

8. Table 5. Why only B/C vs rest? Why not include A? I’m not sure what Tables 5 and 6 add to the results.

Response: We appreciate your feedback, and agree that Tables 5 and 6 do not contribute to the clarification of our results. According to your comment and the editor’s suggestion, we have decided to delete both Tables 5 and 6.

---

## [Decision Letter · Decision Letter 2]

4 Dec 2020

Association between prostate cancer characteristics and BRCA1/2-associated family cancer history in a Japanese cohort

PONE-D-20-24272R2

Dear Dr. Ishiyama,

We’re pleased to inform you that your manuscript has been judged scientifically suitable for publication and will be formally accepted for publication once it meets all outstanding technical requirements.

Kind regards,

Amanda Ewart Toland, Ph.D.

Academic Editor

PLOS ONE

Additional Editor Comments (optional):

Reviewers' comments:

Reviewer's Responses to Questions

**Comments to the Author**

1. If the authors have adequately addressed your comments raised in a previous round of review and you feel that this manuscript is now acceptable for publication, you may indicate that here to bypass the “Comments to the Author” section, enter your conflict of interest statement in the “Confidential to Editor” section, and submit your "Accept" recommendation.

Reviewer #1: All comments have been addressed

Reviewer #2: All comments have been addressed

2. Is the manuscript technically sound, and do the data support the conclusions?

Reviewer #1: Yes

Reviewer #2: Yes

3. Has the statistical analysis been performed appropriately and rigorously? 

Reviewer #1: Yes

Reviewer #2: Yes

4. Have the authors made all data underlying the findings in their manuscript fully available?

Reviewer #1: Yes

Reviewer #2: No

5. Is the manuscript presented in an intelligible fashion and written in standard English?

Reviewer #1: Yes

Reviewer #2: Yes

6. Review Comments to the Author

Reviewer #1: The authors have adequately addressed all comments. The data, though with several limitations, are relevant and timely. Greater understanding of germline variants in Japanese men with PCa is needed, and I recommend that the manuscript be published.

Reviewer #2: The reviewers have addressed all of my remaining concerns. No additional comments or concerns remain.

7. PLOS authors have the option to publish the peer review history of their article (what does this mean?). If published, this will include your full peer review and any attached files.

Reviewer #1: No

Reviewer #2: No

---

## [Editor Report · Acceptance letter]

14 Dec 2020

PONE-D-20-24272R2 

Association between prostate cancer characteristics and *BRCA1/2*-associated family cancer history in a Japanese cohort 

Dear Dr. Ishiyama:

I'm pleased to inform you that your manuscript has been deemed suitable for publication in PLOS ONE. Congratulations! Your manuscript is now with our production department. 

Kind regards, 

on behalf of

Dr. Amanda Ewart Toland 

Academic Editor

PLOS ONE